# Diagnostic Accuracy of Non-Contrast-Enhanced Time-Resolved MR Angiography to Assess Angioarchitectural Classification Features of Brain Arteriovenous Malformations

**DOI:** 10.3390/diagnostics14151656

**Published:** 2024-07-31

**Authors:** Grégoire Chauvet, Mourad Cheddad El Aouni, Elsa Magro, Ophélie Sabardu, Douraied Ben Salem, Jean-Christophe Gentric, Julien Ognard

**Affiliations:** 1Department of Radiology, Hôpital Cavale Blanche, Brest University Hospital, 29200 Brest, France; gregoire.chauvet@chu-brest.fr; 2Department of Interventional Neuroradiology, Hôpital Cavale Blanche, Brest University Hospital, 29200 Brest, France; cheddad.el.aouni@gmail.com (M.C.E.A.); jean-christophe.gentric@chu-brest.fr (J.-C.G.); 3Department of Neurosurgery, Hôpital Cavale Blanche, Brest University Hospital, 29200 Brest, France; elsa.magro@chu-brest.fr; 4Inserm, UMR 1101 (Laboratoire de Traitement de l’Information Médicale-LaTIM), Université de Bretagne Occidentale, 29238 Brest, France; douraied.bensalem@chu-brest.fr; 5Service d’Imagerie Médicale, Hôpital d’Instruction des Armées Legouest, rue des Frères-Lacretelle, 57070 Metz, France; ophelie.sabardu@gmail.com; 6Department of Neuroradiology, Hôpital Cavale Blanche, Brest University Hospital, 29200 Brest, France; 7Inserm, UMR 1304 (GETBO), Western Brittany Thrombosis Study Group, Université de Bretagne Occidentale, 29238 Brest, France

**Keywords:** brain arteriovenous malformations, non-contrast-enhanced MR angiography, contrast-enhanced MR angiography, Spetzler–Martin, buffalo, AVMES, R2DAVM

## Abstract

This study aims to assess the diagnostic accuracy of non-contrast-enhanced 4D MR angiography (NCE-4D-MRA) compared to contrast-enhanced 4D MR angiography (CE-4D-MRA) for the detection and angioarchitectural characterisation of brain arteriovenous malformations (bAVMs). Utilising a retrospective design, we examined 54 MRA pairs from 43 patients with bAVMs, using digital subtraction angiography (DSA) as the reference standard. Both NCE-4D-MRA and CE-4D-MRA were performed using a 3-T MR imaging system. The primary objectives were to evaluate the diagnostic performance of NCE-4D-MRA against CE-4D-MRA and DSA and to assess concordance between imaging modalities in grading bAVMs according to four main scales: Spetzler–Martin, Buffalo, AVM embocure score (AVMES), and R2eDAVM. Our results demonstrated that NCE-4D-MRA had a higher accuracy and specificity compared to CE-4D-MRA (0.85 vs. 0.83 and 95% vs. 85%, respectively) and similar agreement, with DSA detecting shunts in bAVMs or residuals. Concordance in grading bAVMs was substantial between NCE-4D-MRA and DSA, particularly for the Spetzler–Martin and Buffalo scales, with CE-4D-MRA showing slightly higher kappa values for interobserver agreement. The study highlights the potential of NCE-4D-MRA as a diagnostic tool for bAVMs, offering comparable accuracy to CE-4D-MRA while avoiding the risks associated with gadolinium-based contrast agents. The safety profile of imaging techniques is a significant concern in the long-term follow up of bAVMs, and further prospective research should focus on NCE-4D-MRA protocol improvement for clinical use.

## 1. Introduction

Current knowledge on the management of unruptured brain arteriovenous malformations (bAVMs) is limited, and only a few studies reported reliable data. One of those is a 50-month follow-up trial named ARUBA [1]. It demonstrates a superiority of medical management alone in opposition to management with interventional therapy for the prevention of death or symptomatic strokes. The characterisation of angioarchitectural features remains critical in the follow-up of brain arteriovenous malformations (bAVMs), yet today’s imaging methods are limited by radiation exposure and risks associated with contrast agents. Digital subtraction angiography (DSA) continues to be the gold standard for this purpose, offering high sensitivity and specificity. Studies have demonstrated that contrast-enhanced MR angiography (CE-MRA) achieves overall sensitivity and specificity of 0.77 and 0.97, respectively, in detecting residual bAVMs [2]. However, only a few studies have evaluated non-contrast-enhanced MR angiography (NCE-MRA) [2,3,4,5,6], and to our knowledge, only one study has compared NCE-MRA with CE-MRA for bAVM follow-up after gamma knife radiosurgery [7]. Recent advances in magnetic resonance angiography (MRA) have significantly expanded its clinical utility, particularly with the development of NCE 4D MRA. This technique, offering a spatial resolution of 1–1.5 mm^3^ and a temporal resolution of 50–100 ms, has shown promise in characterising cerebrovascular haemodynamics. NCE 4D MRA provides several advantages over DSA and CE-MRA, including its non-invasive nature, which allows for repeated scans in follow-up studies, and its ability to provide detailed haemodynamic and morphological information about cerebral vasculature. Its clinical utility has been evaluated in cases of cerebral malformations and collateral circulations, highlighting its potential as a diagnostic tool. This study aims to compare the diagnostic accuracy of NCE 4D MRA at 3.0 T (4D TRANCE) with CE 4D MRA at 3.0 T (4D TRAK) in detecting residual bAVMs and characterising their angioarchitectural features. By exploring these advanced imaging techniques, we seek to improve the non-invasive diagnostic options available for bAVM follow-up.

## 2. Materials and Methods

### 2.1. Recruitment

Cases were retrospectively assigned from a population of 108 patients followed for a bAVM in our unit from 2015 to 2023. The age of the patients was up to 18 years old. Among them, 57 did not complete an NCE-4D-MRA, and 8 did not complete a DSA in our service. A total of 43 patients were included, with 58 potential NCE-4D-MRA and corresponding CE-4D-MRA and DSA pairs. Four NCE-4D-MRA were uninterpretable, reducing the final study population to 54 MRA pairs (NCE-4D-MRA and CE-4D-MRA within DSA) for 43 patients. Patients had to be previously included in TOBAS, which is an investigator-led, pragmatic, multicentre care trial. The protocol was approved by national review (CPP), and all patients or delegates provided written informed consent [8]. Clinical trial registration number was as follows: NCT02098252.

### 2.2. Four-Dimensional MRA Techniques

Four-dimensional MRA techniques were conducted using a 3-T MR imaging system (Achieva or Ingenia Elition X; Philips Medical System, Best, The Netherlands) for both NCE-4D-MRA and CE-4D-MRA, utilising a commercially available 32-channel head coil. The MR unit’s gradient system achieves a maximum gradient amplitude of 78 mT/m for Ingenia Elition X and 80 mT/m for Achieva, with slew rates of 220 T/m/s and 200 T/m/s, respectively.

The NCE-4D-MRA (Figure 1a–f) was executed using the CINEMA-pASL technique (4D TRANCE; Philips Medical System, Best, The Netherlands) [9], combining STAR angio as an ASL method and look-locker sampling for spin labelling with a 3-dimensional segmented T1-weighted turbo field echo-planar imaging (3D-TI-TFEPI). The sequence involved two acquisitions for each measurement with identical readouts. Control and labelling pulses were applied at the same location. A labelling pulse preceded the first acquisition before the T1 TFEPI readout, followed by a control pulse for the second acquisition. Only spins moving into the imaging volume from flowing blood underwent the labelling pulse. After two acquisitions, temporal phases with identical inversion delays were subtracted, isolating the blood signal and cancelling out the static tissue signal.

The 4D TRANCE sequence’s acquisition parameters included a 3D-T1 TFEPI sequence with TR of 8.1 ms, TE of 4.2 ms, flip angle of 10°, FOV of 210 × 210 mm^2^, matrix of 140 × 129, slab thickness of 150 mm, acquisition voxel of 1.5 × 1.5 × 1.5 mm^3^, reconstruction voxel of 0.73 × 0.73 × 0.75 mm^3^, sensitivity encoding factor of 2.9 in phase-encoding direction, 1.4 in slice-encoding direction, and acquisition time of 5 min, 24 s. The arterial spin labelling volume was 300 mm, with a 20 mm gap between it and the acquisition volume. Ten labelling pulses were applied for dynamic inflow images at intervals of 200 ms (from 200 ms to 2000 ms).

CE-4D-MRA (Figure 1g–l) was performed using the 4D Time-Resolved MR angiography with keyhole (4D-TRAK; Philips Medical System, Best, The Netherlands) sequence [10]. Gadoteric acid (DOTA-Gd, DOTAREM; Guerbert, Villepinte, France) served as the gadolinium-based contrast agent, injected via an automated power injector (Spectris Solaris EP; Medrad Europe, Beek, The Netherlands) at 0.2 mL/kg at 1.5 mL/s, followed by a 25 mL saline flush at the same rate.

The 4D-TRAK technique accelerates image acquisition by combining CENTRA keyhole method, which acquires 15% of the k-space centre in both Ky and Kz directions during the contrast bolus passage, sensitivity encoding (SENSE) for parallel imaging with an acceleration factor of 3.5 in phase-encoding direction and 2.8 in slice-encoding direction, and partial Fourier imaging to skip 20% of the k-space periphery.

The acquisition parameters for the 4D-TRAK sequence were a 3D-T1-EG sequence with TR of 3.7 ms, TE of 1.62 ms, flip angle of 20°, FOV of 240 × 240 mm^2^, matrix of 240 × 209, slab thickness of 180 mm, acquisition voxel of 1 × 1.15 × 2.20 mm^3^, reconstruction voxel of 0.68 × 0.68 × 1.1 mm^3^, and acquisition time of 1 min, 7.8 s. The sequence commenced with a native mask of 4.4 s to subtract stationary tissue, followed by 45 dynamic scans with a temporal resolution of 1400 ms per dynamic scan.

Three-dimensional reconstructions of the 4D-MRAs collected images were systematically conducted on an individual workstation (Viewforum Release, Philips Medical System, Best, The Netherlands), with production of series of dynamic maximum intensity projection (MIP).

### 2.3. DSA Exploration

We utilised 2D Digital Subtraction Angiography (DSA) as the reference standard due to its established gold standard status in bAVM diagnostics. All cerebral catheter angiographies were performed according to the standard of care on a biplanar angiography system (Artis, Siemens Healthineers, Erlangen, Germany). The retained angiogram set consistently included both internal carotid and the dominant vertebral artery images, obtained after a bolus injection of 8 mL of Iodixanol 270 mg/mL (Visipaque 270, GE Healthcare, IDA Business Park, Carrigtwohill, County Cork, Ireland) via a 5F catheter. Frontal and lateral projections were captured at a frequency of 3 images per second. DSA were conducted for diagnostic reasons, either between steps, before a treatment sequence, or to assess cure status, with the interval ranging from less than a day to 1524 days in relation to the 4D-MRAs.

### 2.4. End Points

Initially, our objective was to evaluate the diagnostic accuracy and consistency of NCE-4D-MRA in identifying any pial shunt in patients with bAVMs, whether treated or untreated, with DSA serving as the benchmark for comparison, and to assess how it compares with CE-4D-MRA. Subsequently, our goal was to assess the concordance between these two sequences in relation to the DSA standard for classifying bAVMs according to the four main grading scales: Spetzler–Martin, Buffalo, AVMES, and R2eDAVM, including an analysis of sub-scale items as well as to examine their interobserver agreement.

### 2.5. Interpretation

Image analysis was conducted by three independent readers with varying levels of experience in neuroradiology (10 years, 5 years, and 1 year). All were blinded to each other and to the identifying/clinical variables and treatment status of the participants. Each reader first independently assessed the NCE-4D-MRA set in a random order. After a period of more than three weeks, they evaluated the CE-4D-MRA images in a random order. Interpretations were based solely on the dynamic MIPs provided in three planes. They determined the presence of a shunt in each 4D-MRA and, if identified, proceeded to a standardised descriptive assessment with yielded to the calculation of the four grading scales.

The most experienced reader reviewed the DSA images to verify the clinical data assessments, and discrepancies were resolved through consensus.

Four commonly used scores for classifying bAVMs were selected (Appendix A). These scores describe the architectural characteristics of bAVMs and are routinely used for monitoring and making treatment decisions. The Spetzler–Martin scale, initially developed for assessing surgical risk in bAVM resection, considers the nidus size, bAVM location (eloquent or non-eloquent), and draining veins [11]. The AVM embocure score (AVMES) includes the number of arterial pedicles and demonstrates a predictive rate of complications with the Buffalo grading scale in endovascular embolization [12,13]. A haemorrhagic risk stratification incorporating race, exclusive deep location, AVM size, type of venous drainage, and monoarterial feeding constitutes the R2eDAVM score [14]. Readers were also asked to note the number of venous aneurysms in the reading grid.

### 2.6. Data Analysis

Statistical analysis encompassed descriptive statistics for demographics and selected DSA variables to describe the sample. Four grading scales’ and subitems’ descriptive statistics were also provided (frequencies and percentages; median and interquartile range) for all readings (pooling the 3 readings) with adapted comparison test between 4D-MRAs and DSA. Adapted correlation coefficient and Kappa calculations (for variables with more than two classes, weighted quadratically) were given to detail link between techniques or readings and concordance between techniques/agreement between readers, respectively. It was chosen to consider results of the scales as ordinal variables. κ statistics were interpreted as suggested by Landis and Koch [15]. Performance analysis (sensitivity, specificity, positive predictive value, negative predictive value, and area under the curve) was made for shunt detection for CE-4D-MRA and NCE-4D-MRA compared to the gold standard (DSA) using DeLong method. There were no missing values. Statistical significance was set to 5%. Statistical software used was STATA MP16 Stata Corp, College Station, TX, USA.

## 3. Results

### 3.1. Population

Fifty-four paired examinations (NCE-4D-MRA and CE-4D-MRA) were analysed with corresponding DSA for 43 included patients, who had a median age of 55 years (range, 18–80 years). CE-MRA and NCE-MRA were always conducted simultaneously, and the time between MRA and DSA varied from 0 to 1524 days. Eleven 4D-MRA/DSA pairs from nine patients were recorded at various stages of their disease, including between partial treatment and cure diagnosis. Of the 43 patients, 16 (37.2%) were female, 37 patients (86%) had previously experienced a haemorrhagic event, and 22 (41%) had received treatment. A summary of the sample description can be found in the Table 1. The corresponding flow chart is shown in Figure 2.

### 3.2. Diagnostic Performance and Concordance of 4D-MRAs to Detect Shunts

In the context of a sample of mixed ruptured–unruptured/treated–untreated bAVMs, we recalled 40 visible shunts (74%) on index DSA (29 AVMs and 11 residual shunts). The Nidus size comprised between 0 and 3 cm for 23 AVMs, 3 and 6 cm for 10 AVMs, and up that 6 cm for 7 AVMs. CE-4D-MRA showed a visualisation rate of 64%, while NCE-4D-MRA had a slightly lower rate at 57%. Interobserver agreement, as measured by the kappa statistic, was higher for NCE-4D-MRA (0.82) compared to CE-4D-MRA (0.73). Both modalities demonstrated a similar level of agreement with DSA (0.58–0.59).

The area under the curve (AUC, analysed against DSA) was higher for NCE-4D-MRA at 85%, as opposed to 83% for CE-4D-MRA (*p* = 0.045), indicating a significantly slightly better overall accuracy for the non-contrast technique. The result was mitigated by the sensitivity of CE-4D-MRA that was 81%, which was higher than the 75% sensitivity observed for NCE-4D-MRA. In terms of specificity, NCE-4D-MRA outperformed CE-4D-MRA (95% compared to 85%). The details are reported in Table 2.

### 3.3. Correlation and Concordance of Scaling between DSA and 4D-MRAs

The highest values of correlation and concordance to DSA were obtained with NCE-4D-MRA for the SM grading scale and with CE-4D-MRA for the Buffalo scale.

For the SM grading scale, the correlation coefficients were 0.76 for CE-4D-MRA and 0.78 for NCE-4D-MRA, with kappa values indicating moderate agreement at 0.58 and 0.66, respectively. For the Buffalo scale, the correlation coefficients were 0.79 for CE-4D-MRA and 0.74 for NCE-4D-MRA, while the kappa values showed moderate agreement at 0.65 and 0.58, respectively.

The Figure 3 illustrates the concordance between techniques.

The AVM embocure score or R2eDAVM scales showed strong correlation coefficients, with CE-4D-MRA at 0.80/0.68 and NCE-4D-MRA at 0.73/0.70, but kappa values were the lowest with DSA at 0.57/0.55 and 0.49/0.55, respectively.

The assessments aligned across every variable of the scales except for venous characteristics and the number of feeders. Notably, both correlation and kappa values were above 0.6 for variables such as nidus size, the presence of a deep location, the eloquence of location, and the size of the feeder. The agreement was slightly lower for the number of feeders. The description of venous characteristics was notably weaker with kappa values ranging from 0.24 to 0.46 for CE-4D-MRA and from 0.16 to 0.20 for NCE-4D-MRA. The intermodality analysis for all readings is detailed in Table 3.

### 3.4. Agreements of Readings 4D-MRAs Techniques

The results are summarised in Table 4.

The kappa values indicated a range from moderate to substantial agreement for the SM and Buffalo grading scales. Specifically, CE-4D-MRA demonstrated kappa values of 0.67 for SM and 0.70 for Buffalo, indicating a good level of agreement among readers. NCE-4D-MRA showed slightly lower kappa values of 0.54 for SM and 0.59 for Buffalo.

The AVMES score revealed a kappa of 0.69 for CE-4D-MRA and 0.60 for NCE-4D-MRA, suggesting a moderate agreement. The R2eDAVM scale had the lowest kappa values, with 0.29 for CE-4D-MRA and 0.38 for NCE-4D-MRA, indicating a fair level of agreement.

Except for the assessment of R2DAVM (which had the lowest interobserver agreement whatever the chosen technique), the interobserver agreement (and the correlation coefficient) was always slightly higher for CE-4D-MRA than NCE-4D-MRA.

When comparing NCE-4D-MRA to CE-4D-MRA, kappa values for SM, AVMES, Buffalo, and R2eDAVM ranged from 0.48 to 0.55.

## 4. Discussion

This study evaluated the diagnostic accuracy of NCE-4D-MRA at 3.0 Tesla for detecting shunts in patients with bAVMs, whether treated/cured or not. The sensitivity and specificity of NCE-4D-MRA were found to be 95% and 75%, respectively. Notably, there was a similar level of agreement with CE-4D-MRA compared to DSA. The slight superior diagnostic accuracy of NCE-4D-MRA may be attributed to its enhanced spatial and temporal resolution, which boasts a reconstruction voxel size of 0.73 × 0.73 × 0.75 mm^3^ and a temporal resolution of 0.20 s per volume compared to CE-4D-MRA’s voxel size of 0.68 × 0.68 × 1.1 mm^3^ and temporal resolution of 1.4 s per volume. The differences in temporal resolution can be seen in Figure 1.

While the study showcases the commendable diagnostic accuracy of NCE-4D-MRA in evaluating bAVMs, its inherent limitations caution against unequivocally ruling out the bAVM cure based solely on a negative NCE-4D-MRA result. The high sensitivity and positive predictive value underscore NCE-4D-MRA’s reliability in positive cases, virtually eliminating the risk of false positives. The utilisation of NCE-4D-MRA has also been assessed in radiosurgery planning, with an association of CE-4D-MRA and TOF [16] or with the ASL technique alone [17]. In a study of radiosurgically treated patients with small brain AVMs, the couple arterial spin-labelling/TOF was found to be superior to gadolinium-enhanced MR imaging in detecting residual bAVMs [18]. When current recommendations on follow-ups consider DSA as the gold standard for the detection of residual or recurrent AVMs after the apparent cure of bAVMs [19], NCE4D-MRA’s precision could facilitate the reduction of unnecessary DSA exams, particularly beneficial for patients under surveillance post-radiosurgery.

Our study also evaluated the consistency of NCE-4D-MRA in grading the four most frequently utilised scores relative to DSA. Concordances of NCE-4D-MRA with DSA were found substantial for Spetzler–Martin and Buffalo (kappa of 0.66 and 0.58, respectively); and found in the same range that CE-4D-MRA (kappa of 0.58 and 0.65, respectively). A wide range of quotation for R2DAVM and AVMES scores might account for lower concordance and correlation with DSA for our two MRA modalities, as it increases the risk of varying quotation. Considering these two scales, interobserver agreement always revealed slightly higher kappa values for CE-4D-MRA (SM: 0.67 vs. 0.53; Buffalo: 0.70 vs. 0.55). An analysis of the sub-scales’ detailed features revealed a decline in the efficacy of NCE-4D-MRA compared to CE-4D-MRA in venous drainage characteristics (Figure 1 and Figure 2), in particular demonstrating limited intermodality and interobserver agreements, likely due to the diminishing subtracted signal from the T1 recovery of magnetically labelled blood. Significant advancements have been achieved since the last decade and Nakamura et al. [9] for the T1 weighted relaxation time. Iryo et al. [4] described a significant detection of venous drainage patterns (intermodality agreement κ = 0.88 and interobserver agreement κ = 0.86) with a T1 weighted relaxation time of 8.5 s. This suggests that optimising the T1 relaxation time could be a promising direction for enhancing the depiction of venous drainage in ASL-based MRA. Fei Cong et al. [20] prolonged the longitudinal relaxation time at 7 Tesla, which provides a longer time window to collect the ASL signal and increases the analysis of veinous characteristics. Reflection can also consider the length of the echo time (TE). Some authors employ an ultrashort TE to minimise the phase dispersion of the labelled blood flow signal within the voxel space and mitigate magnetic susceptibility effects [21,22]. A short TI reconstruction enables even higher temporal resolution, albeit with certain limitations, such as fat-related artifacts and the reduced visibility of distal arteries [23].

Temporal resolution can also be controlled through the spin labelling technique, such as super-selective pseudo-continuous arterial spin labelling combined with CENTRA-keyhole and view sharing (4D-S-PACK), which can be useful for visualising bAVMs [24]. Murazaki et al. [25] introduced two additional strategies to improve temporal resolution and decrease scanning times in the latter ASL technique: compressed sensitivity encoding (CS-SENSE) and PhyZiodynamics. These methods facilitate a more comprehensive analysis of inflow feeding arteries without extending sequence duration. A prospective study evaluated selective arterial spin labelling in addition to CE-4D-MRA for the anatomic and functional characterisation of bAVMs in the comparison of DSA. Corresponding Spetzler–Martin grading and venous drainage patterns (deep versus superficial) were described similarly (100% of match) [26]. The principles and clinical applications of the latest ASL-based MRA techniques were reviewed in a recent study [27].

Despite a demonstrated strength of CE-4D-MRA to classify bAVMs and evaluate their haemodynamic features [28,29], its exposure to gadolinium-based contrast agents (GBCAs) is significant [30]. Those risks are mostly represented by gadolinium brain deposition and nephrotoxicity [31,32,33,34,35,36]. Furthermore, immediate adverse reactions to gadolinium-based agents can be observed, and there is still a lack of data over GBCA use during pregnancy and lactation [37]. In the absence of definitive data, one should contemplate the use of alternative diagnostic methods, like non-contrast MRI [38]. Employing non-contrast-enhanced sequences may mitigate the risks associated with GBCAs and may streamline the imaging process for paediatric patients.

BAVM cares need consistent information about arterial feeders and draining vein features [39]. Therapy and follow-up decisions are conditioned by the reliability of the imaging technique [19]. Ongoing bAVM management strategies might rely on a noninterventionist follow-up. In TOBAS [40], Raymond et al. revealed the infrequent success of endovascular embolization alone, with treatment efficacy defined as the complete occlusion of the bAVM on follow-up catheters or non-invasive angiography, at under 40%. In a randomised trial, Darsaut et al. established that surgical management provides a high curative level at 88% of all patients with unruptured or ruptured bAVMs but with a risk of disabling complications of 4% [41].

Due to its superior ability to identify small-size arterial feeders and draining vein features, DSA continues to serve as the gold standard and is imperative for interventional planning procedures in brain arteriovenous malformations [42] (Figure 2). However, 4D-MRA serves as a valuable component of medical follow-up strategies.

Whilst association among bAVM blood flow and rupture risk is suggested [43], ASL-4D-MRA could also offer additional information by monitoring the haemodynamic flow of bAVMs, which could be linked to predicting haemorrhagic risk [44].

Our findings demonstrated an equivalence of NCE-4D-MRA with CE-4D-MRA in the graduation of Spetzler–Martin, Buffalo, and AVMES scores. Moreover, it provides a moderate-to-substantial agreement for the feeder number count (intermodality agreement r = 0.728, κ = 0.559; interobserver agreement r = 0.821, κ = 0.649). Particularly for this chronic pathology that requires long-term monitoring, this might suggest a follow-up procedure without the use of gadolinium-based agents. These findings underscore the importance of patient-centred care in managing bAVMs: given the chronic nature of bAVMs and the potential need for lifelong surveillance, the safety profile of imaging techniques becomes a paramount concern.

To the best of our knowledge, this study is the inaugural comparison of the diagnostic accuracy and concordance between DSA and non-contrast-enhanced 4D MRA versus contrast-enhanced 4D MRA in bAVM shunt detection and grading involving a population of this size. Xu et al. [45] reported fifteen bAVMs detected in DSA explored in NCE-4D-MRA: intermodality agreements were excellent for the arterial feeders (Kappa of 0.93), good for the nidus size (0.69), and moderate for the venous drainage (0.49), in accordance with our findings.

Despite its strengths, the study acknowledges limitations, such as the retrospective design, the variable intervals between 4D-MRA and DSA, and the potential for angioarchitectural changes over time. In our investigation, twenty-two patients received treatment, predominantly endovascular (either partial or complete), and six also underwent radiosurgery, which incrementally diminished the elevated blood flow over a span of 2 to 5 years and potentially led to its elimination [28,46]. This possibly led to changes in bAMV features. A comprehensive study conducted by Soize et al. [47] demonstrates comparable concordance and correlation for detecting residual shunts in treated bAVMs with CE-4D-MRA compared to DSA. The sensitivity was 73.7%, specificity 100%, positive predictive value 100%, negative predictive value 78.3%, with a moderate intermodality agreement (kappa = 0.60).

Another limitation of our study was the lack of sound-to-noise ratio (SNR) and contrast-to-noise ratio (CNR) measurement and comparison of those between NCE-4D-MRA and CE-4D-MRA. The SNR and CNR are indeed great parameters for assessed dynamic, morphologic, and functional information about bAVMs. Günther et al. demonstrated the implication of SNR in ASL brain perfusion and how it can be quantified and adjusted with flip angle changes [48]. The ASL technique in NCE-4D-MRA can also deliver an angiographic depiction of intracranial vessels with precision and high CNR [49]. As a CNR analysis can be performed in different vessel sections and can be used to compare two sequences [9], it could be employed in further studies to compare NCE-4D-MRA and CE-4D-MRA.

Future research should focus on prospective studies with tighter imaging intervals. The refinement of NCE-4D-MRA protocols could lead to enhance venous detail visualisation. Lastly, the exploration of artificial intelligence algorithms for automated grading and assessment of bAVMs might assist in reducing the interobserver factor in the reading [50]. Other ways of grading the bAVM may also be evaluated regarding the aim of the follow-up [51], and NCE methods of follow up may be developed using a combined sequence analysis, relying, for example, on SWI [52], TOF, and ASL [53] to find the optimised protocol in a desired subset of follow-up.

Moreover, there is a need to assess the long-term outcomes and impact of bAVM patients managed based on NCE (-4D-) MRA findings, especially in comparison to traditional management pathways.

## 5. Conclusions

This study reports encouraging results for the use of contrast-free 4D MRA techniques in the detection of shunts in bAVMs and their characterisation compared with 4D MRA with gadolinium injection, with cerebral arteriography as the reference. Many improvements can be made to this sequence, which would avoid repeated exposure to X-rays and contrast agents for patients with bAVMs requiring long-term follow-up.

## Figures and Tables

**Figure 1 diagnostics-14-01656-f001:**
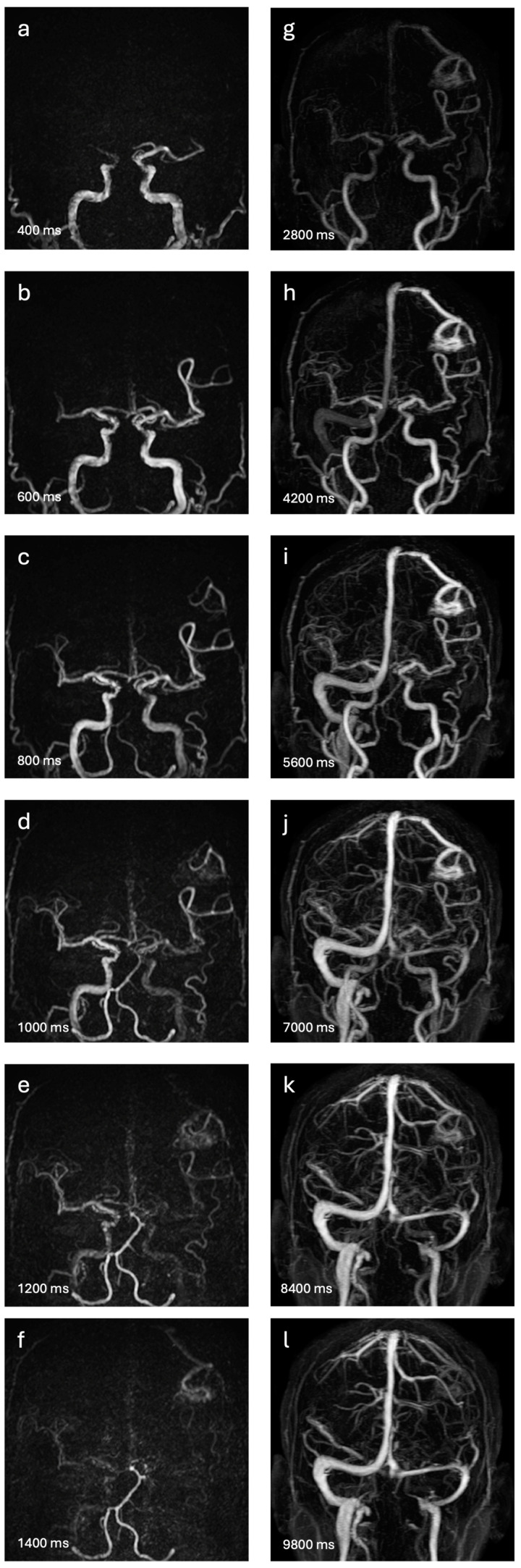
Magnetic resonance angiography of a thirty-five-year-old man with an untreated frontoparietal AVM, initially discovered on a haemorrhagic event. The anteroposterior maximal intensity projections of both 4D-MRA techniques studied are presented. The six most relevant of ten dynamic images of NCE-4D-MRA appear in (**a**–**f**), with a temporal resolution of 200 ms. Corresponding dynamic images of CE-4D-MRA appear in (**g**–**l**), with a temporal resolution of 1400 ms. The images reveal two arterial feeders from the left middle cerebral artery, a small nidus (<30 mm) in the postcentral gyrus, and a superficial veinous drainage in the superior sagittal sinus. The temporal resolution of NCE-4D-MRA allows for precision in the arterial analysis, but the loss of signal in the last scan times limits veinous characteristic descriptions.

**Figure 2 diagnostics-14-01656-f002:**
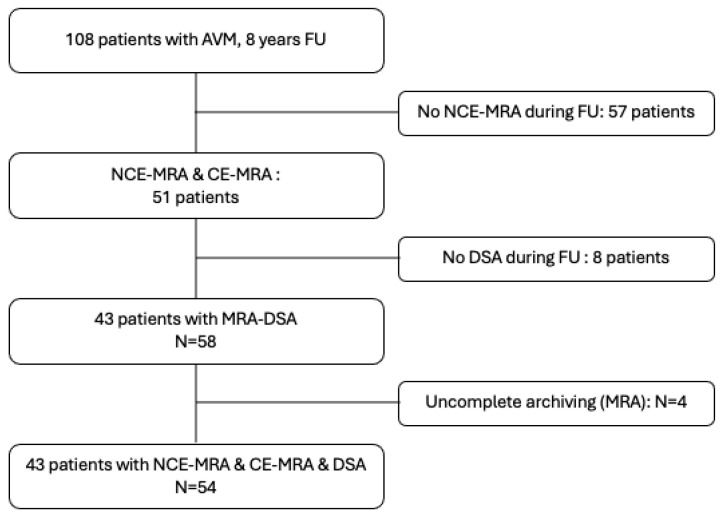
Flow chart.

**Figure 3 diagnostics-14-01656-f003:**
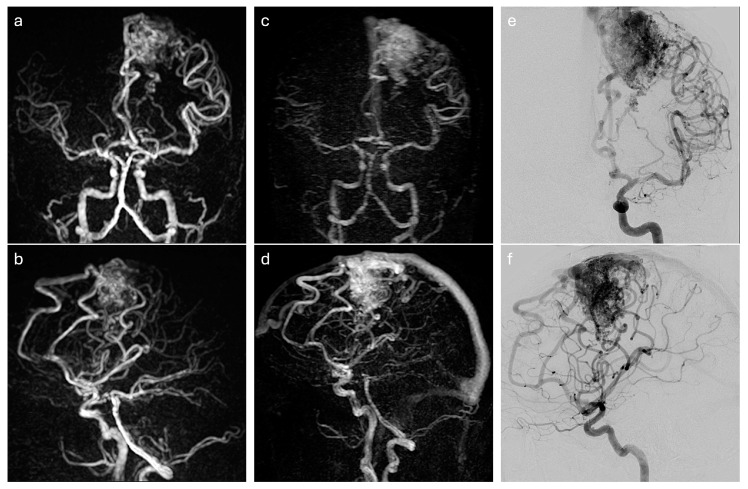
An illustration of a forty-year-old man with an unruptured and untreated brain arteriovenous malformation (bAVM). The anteroposterior (**a**) and lateral (**b**) maximal intensity projection (MIP) of a non-contrast-enhanced four-dimensional magnetic resonance angiography (NCE-4D-MRA) at the best nidus phase. The anteroposterior (**c**) and lateral (**d**) maximal intensity projection (MIP) of a contrast-enhanced four-dimensional magnetic resonance angiography (CE-4D-MRA) at the best nidus phase. The anteroposterior (**e**) and lateral (**f**) projections of a digital subtraction angiography (DSA) at the best nidus phase. The images show a bAVM in eloquence localisation (precentral gyrus in left frontal lobe), with a nidus size between 30 mm and 60 mm, six arterial feeders from the left anterior artery and left middle cerebral artery, and a superficial veinous drainage in the superior sagittal sinus. NCE-4D-MRA allows for precision about arterial feeders with a lack of signal on veinous drainage, while the temporal resolution of CE-4D-MRA does not disassociate arterial and venous phases as accurately as NCE-4D-MRA and DSA. DSA outperforms 4D-MRAs in an arterial feeder analysis.

**Table 1 diagnostics-14-01656-t001:** Patient demographic, treatment, and grading scale information.

Description	Med. or Freq.	[IQR] or %
Demographics (over *n* = 43)		
Age, years	55	[34–61]
Sex, male	27	62.80%
Hemorrhagic presentation	37	68.52%
Treated (overall *n* = 54)	22	40.74%
Embolization	8	36.36%
Embolization then radiosurgery	6	27.27%
Embolization then surgery	5	22.73%
Radiosurgery	3	13.64%
bAVM location (over *n* = 43)		
Supratentorial	38	88.30%
Infratentorial	5	11.63%
bAVM visualisation (DSA consensus)	40	74.07%
Shunt (over non-treated *n* = 32)	29	90.62%
Residual shunt (over treated *n* = 22)	11	50.00%
bAVM Grading (DSA consensus, overall)		
SM	1	[1–3]
AVMES	3	[3–6]
Buffalo	2	[2–3]
R2DAVM	4	[4–5]
Time intervals (overall)		
Delay between DSA and 4D MRA, months	4	[1–18]

Note: *n* = Number; Med. = median and Freq. = frequency; IQR = interquartile range; DSA = digital subtraction angiography; bAVM = brain arteriovenous malformation; SM = Spetzler–Martin; AVMES = arteriovenous malformation embocure score; R2DAVM = race, exclusive deep, arteriovenous malformation; 4D MRA = four-dimensional magnetic resonance angiography.

**Table 2 diagnostics-14-01656-t002:** Diagnostic performance and concordance of contrast-enhanced four-dimensional magnetic resonance angiography (CE-4D-MRA) and non-contrast-enhanced four-dimensional magnetic resonance angiography (NCE-4D-MRA) in shunt detection.

Shunt Detection	CE-4D-MRA	NCE-4D-MRA
Visualisation (%)	63.58	56.79
Kappa (interobs.)	0.73	0.82
Kappa (vs. DSA)	0.59	0.58
Sensitivity	80.83	75.00
Specificity	85.71	95.24
PPV	84.17	97.82
NPV	61.10	57.14
AUC *	83.27	85.11

Note: interobs. = interobserver; PPV = predictive positive value; NPV = negative predictive value; AUC = area under the curve. * Sig *p* = 0.0452

**Table 3 diagnostics-14-01656-t003:** Intermodality agreement between contrast-enhanced four-dimensional magnetic resonance angiography (CE-4D-MRA) and digital subtraction angiography (DSA) for all readings; intermodality agreement between non-contrast-enhanced 4D MRA (NCE-4D-MRA) and DSA for all readings.

	CE-4D-MRA (vs. DSA)					NCE-4D-MRA (vs. DSA)				
Variables	Med./Freq.	IQR/%	*p*-Value	Corr. Coef.	Kappa	Med./Freq.	IQR/%	*p*-Value	Corr. Coef.	Kappa
*Grading scales*										
SM	2	[1–4]	1.000	0.76	0.58	3	[1–4]	<0.001 *	0.78	0.66
AVMES	4	[3–7]	0.007 *	0.80	0.57	4	[3–6]	0.014 *	0.73	0.49
Buffalo	2	[2–4]	0.714	0.79	0.65	3	[2–3.25]	0.357	0.74	0.58
R2DAVM	4	[3–5]	0.648	0.68	0.55	4.5	[4–5]	0.034 *	0.70	0.55
*Features*										
Nidus size (SM)	1	[1–2]	0.555	0.85	0.77	2	[1–2]	0.193	0.87	0.80
Deep location	14	08.64%	<0.001 *	0.83	0.82	16	09.87%	<0.001 *	0.67	0.66
Eloquency	55	33.95%	0.021 *	0.70	0.69	56	34.56%	0.021 *	0.71	0.70
Fossa posterior	13	08.02%	<0.001 *	0.96	0.95	10	06.17%	<0.001 *	1.00	1.00
Nb. of feeders	3	[1.5–5]	0.173	0.86	0.55	3	[2–5]	0.315	0.78	0.47
Feeder size > 1 mm	88	90.72	<0.001 *	0.71	0.71	81	90.00	<0.001 *	0.60	0.59
Superf. ven. drain.	61	59.80%	<0.001 *	0.54	0.46	69	76.67	0.019 *	0.20	0.16
Nb. draining veins	1	[1–2]	0.101	0.48	0.44	1	[1–1]	<0.001 *	0.30	0.20
Nb. ven. aneurysm	0	[0–0.5]	<0.001 *	0.27	0.24	0	0	<0.001 *	0.31	0.17

Note: Med. = median and Freq. = frequency; IQR = interquartile range; Corr. Coef = correlation coefficient, SM = Spetzler–Martin; AVMES = arteriovenous malformation embocure score; R2DAVM = race, exclusive deep, arteriovenous malformation; Nb. = number; Ven. = veinous; * statistically significant.

**Table 4 diagnostics-14-01656-t004:** Interobserver (interobs.) agreement of contrast-enhanced four-dimensional magnetic resonance angiography (CE-4D-MRA) for all readings. Interobserver agreement of non-contrast-enhanced four-dimensional magnetic resonance angiography (NCE-4D-MRA) for all readings. Intermodality agreement comparison of CE-4D-MRA and NCE-4D-MRA for all readings.

Agreements between 4D-MRAs Techniques						
Variables	CE-4D-MRA (Interobs.)		NCE-4D-MRA (Interobs.)		NCE- vs. CE-4D-MRA	
	Corr. Coeff.	Kappa	Corr. Coeff.	Kappa	Corr. Coeff.	Kappa
*Grading scales*						
SM	0.84	0.67	0.74	0.54	0.68	0.53
AVMES	0.87	0.69	0.80	0.60	0.68	0.49
Buffalo	0.83	0.70	0.79	0.59	0.70	0.55
R2DAVM	0.52	0.29	0.60	0.38	0.60	0.48
*Features*						
Nidus size (SM)	0.90	0.82	0.85	0.76	0.80	0.71
Deep location	0.80	0.68	0.69	0.44	0.58	0.58
Eloquency	0.65	0.46	0.57	0.33	0.51	0.51
Fossa posterior	1.00	1.00	1.00	1.00	1.00	1.00
Nb. of feeders	0.92	0.79	0.82	0.65	0.73	0.56
Feeder size > 1 mm	0.47	0.19	-	0.03	0.40	0.39
Superf. ven. drain.	0.64	0.40	0.44	0.13	0.17	0.21
Nb. draining veins	0.79	0.35	0.46	0.10	0.28	0.21
Nb. ven. aneurysm	0.47	0.11	-	0.14	0.27	0.23

Note: Corr. Coef = correlation coefficient, SM = Spetzler–Martin; Nb. = number; superf; = superficial; ven. = veinous; drain. = drainage.

## Data Availability

Data available on request due to restrictions (e.g., privacy, legal or ethical reasons).

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
