# Peer review of "Diagnostic Accuracy of Non-Contrast-Enhanced Time-Resolved MR Angiography to Assess Angioarchitectural Classification Features of Brain Arteriovenous Malformations"

_diagnostics, 2024, doi:10.3390/diagnostics14151656_

Round 1
Reviewer 1 Report
Comments and Suggestions for Authors
The article evaluates the accuracy of 4D TRAK sequence in bAVM, and compares it with that of CE MRA and DSA. It is well-stractured and fluent.
Author Response
Thank you for your reviewing and your comment.
Reviewer 2 Report
Comments and Suggestions for Authors
The paper is very well written and I believe it is suitable for publication in its present form.
Author Response

(The authors gave the same response as above.)

Reviewer 3 Report
Comments and Suggestions for Authors
Major comments that need to be addressed in a revised manuscript
1. This is a well-structured study that assessed the diagnostic accuracy of non-contrast-enhanced 4D MR angiography (NCE-4D-MRA) compared to contrast-enhanced 4D MR angiography (CE-4D-MRA) for detecting and characterizing brain AVMs. It is, however, noted that some portions are difficult to understand. A brush-up in English expression would be desirable, especially regarding readability throughout the manuscript. Additionally, some revisions are mandatory regarding minor issues as specifically stated below.
Specific comments
Abstract
2. The last two sentences are prose-like and should be rewritten.
Introduction
3. The background of the need for the NCE-4D-MRA in diagnosing brain AVMs is little mentioned. Shortage of previous studies in the same area is insufficient. Why the NCE-4D-MRA is to be considered in the radiological evaluation of brain AVMs? Please clarify.
Materials and Methods
4. Is “Ingenia Edition X” correct? Isn’t it “Ingenia Elition X”?
Results
5. Please reconfirm the number of patients. The authors wrote “Of the 43 patients, 20 (37%) were female” in the text. However, in Table 1, the number of male patients is 22.
6. In 3.3. the authors state “The highest values of correlation and concordance to DSA were obtained with NCE-4D-MRA for the SM grading scale and with CE-4D-MRA for the Buffalo scale”. Corresponding discussion regarding the reason and/or clinical significance seems lacking in Discussion. This is to be sufficiently discussed.
Discussion
7. As a limitation of this study, it did not comparatively assess the clarity of three major components of AVMs as measured by SNR or CNR on NCE-4D-MRA and CE-4D-MRA. This needs to be fully discussed.
Figures and Tables
8. Figure sub numbers (A, B, C….) in Figs. 1 and 3 should be changed to lower-case letters as in the text and legends.
Comments on the Quality of English LanguageA brush-up in English expression would be desirable, especially regarding readability throughout the manuscript.
Author Response
Thank you for your review and constructive comments. Please find the detailed responses below and the corresponding revisions/corrections highlighted/in track changes in the re-submitted files.
Comment 1: This is a well-structured study that assessed the diagnostic accuracy of non-contrast-enhanced 4D MR angiography (NCE-4D-MRA) compared to contrast-enhanced 4D MR angiography (CE-4D-MRA) for detecting and characterizing brain AVMs. It is, however, noted that some portions are difficult to understand. A brush-up in English expression would be desirable, especially regarding readability throughout the manuscript. Additionally, some revisions are mandatory regarding minor issues as specifically stated below.
Response 1: We've corrected a few long sentences in the text and reworded it. One of our English native acquaintance helped us to point it and to highlight writing errors like the wrong use of plurals sometimes.
Comment 2: The last two sentences are prose-like and should be rewritten.
Response 2: We rewritted it like:
The safety profile of imaging techniques is a significant concern in the long-term follow up of bAVMs and further prospective research should focus on NCE-4D-MRA protocol improvement for clinical use.
Comment 3: The background of the need for the NCE-4D-MRA in diagnosing brain AVMs is little mentioned. Shortage of previous studies in the same area is insufficient. Why the NCE-4D-MRA is to be considered in the radiological evaluation of brain AVMs? Please clarify.
Response 3: We rewritted the introduction and inserted it at line 56 (in manuscript.v2) like:
The characterization of angioarchitectural features remains critical in the follow-up of brain arteriovenous malformations (bAVMs), yet today's imaging methods are limited by radiation exposure and risks associated with contrast agents. Digital subtraction angiography (DSA) continues to be the gold standard for this purpose, offering high sensitivity and specificity. Studies have demonstrated that contrast-enhanced MR angiography (CE-MRA) achieves overall sensitivity and specificity of 0.77 and 0.97, respectively, in detecting residual bAVM (2). However, only a few studies have evaluated non-contrast-enhanced MR angiography (NCE-MRA) (3–6,6,7), and to our knowledge, only one study has compared NCE-MRA with CE-MRA for bAVM follow-up after gamma knife radiosurgery (8). Recent advances in magnetic resonance angiography (MRA) have significantly expanded its clinical utility, particularly with the development of NCE 4D MRA. This technique, offering a spatial resolution of 1–1.5 mm³ and a temporal resolution of 50–100 milliseconds, has shown promise in characterizing cerebrovascular hemodynamics. NCE 4D MRA provides several advantages over DSA and CE-MRA, including its noninvasive nature, which allows for repeated scans in follow-up studies, and its ability to provide detailed hemodynamic and morphological information about cerebral vasculature. Its clinical utility has been evaluated in cases of cerebral malformations and collateral circulations, highlighting its potential as a diagnostic tool. This study aims to compare the diagnostic accuracy of NCE 4D MRA at 3.0 T (4D TRANCE) with CE 4D MRA at 3.0 T (4D TRAK) in detecting residual bAVMs and characterizing their angioarchitectural features. By exploring these advanced imaging techniques, we seek to improve the noninvasive diagnostic options available for bAVM follow-up.
Comment 4: Is “Ingenia Edition X” correct? Isn’t it “Ingenia Elition X”?
Response 4: Exactly. We corrected it.
Comment 5: Please reconfirm the number of patients. The authors wrote “Of the 43 patients, 20 (37%) were female” in the text. However, in Table 1, the number of male patients is 22.
Response 5: Our mistake, we reconfirmed the population and above the 43 patients there are 27 males (62.8%) and 16 females (37.2%).
Comment 6: In 3.3. the authors state “The highest values of correlation and concordance to DSA were obtained with NCE-4D-MRA for the SM grading scale and with CE-4D-MRA for the Buffalo scale”. Corresponding discussion regarding the reason and/or clinical significance seems lacking in Discussion. This is to be sufficiently discussed.
Response 6: We add a sentence in the discussion and inserted it at line 312 (in manuscript.v2) like:
A wide range of quotation for R2DAVM and AVMES scores might account for the lower concordance and correlation with DSA for our two MRA modalities, as it increases the risk of varying quotation.
Comment 7: As a limitation of this study, it did not comparatively assess the clarity of three major components of AVMs as measured by SNR or CNR on NCE-4D-MRA and CE-4D-MRA. This needs to be fully discussed.
Response 7: Thank you for pointing this out. We add a paragraph in the discussion and inserted it at line 397 (in manuscript.v2) like:
Another limitation of our study was the lack of sound-to-noise ratio (SNR) and contrast-to-noise ratio (CNR) measurement and comparison of those between NCE-4D-MRA and CE-4D-MRA. The SNR and CNR are indeed great parameters for assessed dynamic, morphologic and functional information about bAVMs. Günther and al. demonstrated the implication of SNR in ASL brain perfusion, how it can be quantified and adjusted with flip angle changes (1). The ASL technique in NCE-4D-MRA can also deliver an angiographic depiction of intracranial vessels with precision and high CNR (2). As CNR analysis can be performed in different vessel sections and can be used to compare two sequences (3), it could be employed in further studies to compare NCE-4D-MRA and CE-4D-MRA.
- Günther M, Bock M, Schad LR. Arterial spin labeling in combination with a look‐locker sampling strategy: Inflow turbo‐sampling EPI‐FAIR (ITS‐FAIR). Magnetic Resonance in Med. nov 2001;46(5):974‑84.
- Jagadeesan B, Tariq F, Nada A, Bhatti IA, Masood K, Siddiq F. Principles Behind 4D Time-Resolved MRA/Dynamic MRA in Neurovascular Imaging. Seminars in Roentgenology. avr 2024;59(2):191‑202.
- Nakamura M, Yoneyama M, Tabuchi T, Takemura A, Obara M, Tatsuno S, et al. Vessel-selective, non-contrast enhanced, time-resolved MR angiography with vessel-selective arterial spin labeling technique (CINEMA–SELECT) in intracranial arteries. Radiol Phys Technol. juill 2013;6(2):327‑34.
Comment 8: Figure sub numbers (A, B, C….) in Figs. 1 and 3 should be changed to lower-case letters as in the text and legends.
Response 8: We agree. We changed it figures 1 and 3.
Thank you again for your time and your valuable comments.